# Induction of resident memory T cells enhances the efficacy of cancer vaccine

Mevyn Nizard[1,2], Hélène Roussel[1,2,3,*], Mariana O. Diniz[1,4,*], Soumaya Karaki[1,2], Thi Tran[1,2], Thibault Voron[1,2], Estelle Dransart[5], Federico Sandoval[1,2], Marc Riquet[6], Bastien Rance[7], Elie Marcheteau[1,2], Elizabeth Fabre[8], Marion Mandavit[1,2], Magali Terme[1,2], Charlotte Blanc[1,2], Jean-Baptiste Escudie[7], Laure Gibault[3], Françoise Le Pimpec Barthes[6], Clemence Granier[1,2], Luis C.S. Ferreira[4], Cecile Badoual[1,2,3], Ludger Johannes[5] & Eric Tartour[1,2,3]

Tissue-resident memory T cells (Trm) represent a new subset of long-lived memory T cells that remain in tissue and do not recirculate. Although they are considered as early immune effectors in infectious diseases, their role in cancer immunosurveillance remains unknown. In a preclinical model of head and neck cancer, we show that intranasal vaccination with a mucosal vector, the B subunit of Shiga toxin, induces local Trm and inhibits tumour growth. As Trm do not recirculate, we demonstrate their crucial role in the efficacy of cancer vaccine with parabiosis experiments. Blockade of TFGβ decreases the induction of Trm after mucosal vaccine immunization, resulting in the lower efficacy of cancer vaccine. In order to extrapolate this role of Trm in humans, we show that the number of Trm correlates with a better overall survival in lung cancer in multivariate analysis. The induction of Trm may represent a new surrogate biomarker for the efficacy of cancer vaccine. This study also argues for the development of vaccine strategies designed to elicit them.

[1] INSERM U970, Université Paris Descartes, Sorbonne Paris-Cité, 56 Rue Leblanc, Paris 75015, France. [2] Equipe Labellisée Ligue Contre le Cancer, Paris 75015, France. [3] Department of Pathology, Hopital Européen Georges Pompidou, 20 Rue Leblanc, Paris 75015, France. [4] Institute of Biomedical Sciences, University of Sao Paulo, Av Prof Lineu Prestes, Sao Paulo SP-CEP 05508-900, Brazil. [5] Institut Curie, PSL Research University, Chemical Biology of Membranes and Therapeutic Delivery Unit, INSERM U 1143, CNRS UMR3666, 26 Rue d'Ulm 75248, Paris Cedex 05, France. [6] Hopital Europeen Georges Pompidou, Chrirurgie Thoracique Générale, Oncologique et Transplantation, 20 Rue Leblanc, Paris 75015, France. [7] Department of Medical Bioinformatics, Hopital Européen Georges Pompidou, 20 Rue Leblanc, Paris 75015, France. [8] Departement of Medical Oncology, Hopital Européen Georges Pompidou, 20 Rue Leblanc, Paris 75015, France. * These authors contributed equally to this work. Correspondence and requests for materials should be addressed to E.T. (email: eric.tartour@aphp.fr).

Resident memory T cells (Trm) represent a new subset of long-lived memory T cells that remain in tissue and do not recirculate. They are found most prominently at mucosal sites in contact with the environment (lung, digestive and genital tract) and skin[1–3]. They are transcriptionally, phenotypically and functionally distinct from other recirculating T-cell subsets, such as central memory and effector memory T cells. Trm cells were derived from the same KLRG1[−] precursor T cells that gave rise to the long-lived memory cells found in the circulation[4]. Some core genes (*Itga1* (integrin α1: CD49a), *Itgae* (integrin αE: CD103), *CD69*, *cdh1* (E-cadherin)) expressed by most Trm located in various organs were identified[2,4–6]. However, Trm from various tissues may differentially express specific adhesion molecules and chemokine receptors favouring their retention to different tissues[4]. *In vitro*, TGFβ combined with IL-33 and TNFα can induce a Trm cell-like phenotype defined by the expression of CD103 and CD69 (ref. 7).

In many infectious models both circulating effector T cells and Trm were induced, but in the absence of Trm mice were only suboptimally protected[8–12]. Trm cells are highly protective during localized infections and display the capacity to (i) directly kill pathogen-infected cells, and (ii) release cytokines and chemokines amplifying local recruitment of other innate and adaptive immune cells[8,9,13]. The presence of Trm in the mouse lungs has been described as a better surrogate marker, than memory-specific T cells in the blood, for the protection against reinfection in different preclinical studies in mice[14]. In non-human primates, the presence of Trm against SIV or Ebola virus was essential to control the viral load[15,16].

In contrast to their well-defined role in infectious models, data on Trm in cancer are just emerging. In a preclinical model of spontaneous breast cancer, a natural immune response composed of resident innate lymphoid cells close to ILC1 and TCR (αβ and γδ) positive T cells was described to occur. These cells expressed CD103 and CD49a, did not recirculate and delayed tumour growth[17]. In humans, CD103[+]CD8[+]T cells were present in the epithelial region of ovarian cancer, urothelial cell carcinoma and lung carcinoma. Trm also mediates specific cytolytic activity towards autologous human tumour cells[18].

Due to their role in local immunity, strategies to elicit Trm after vaccination have been developed. We and others clearly have shown that mucosal immunization was more efficient than the conventional systemic route (intramuscular, subcutaneous) to elicit Trm at the mucosal tumour site[19–22]. Indeed, the mucosal route of immunization imprints T cells with a mucosal homing programme defined by a profile of integrin and chemokine receptors promoting their homing to the site of initial activation[23]. A correlation was observed between the ability to elicit these cells at the tumour site and the control of tumour growth[24]. However, up until now a direct demonstration of the role of Trm in cancer immunosurveillance after cancer vaccine administration is lacking.

We thus thoroughly characterize Trm after mucosal cancer vaccine administration and in human lung cancer. Using various approaches to discriminate the respective role of effector memory T cells and Trm, this study clearly demonstrates that Trm play a key role in the efficacy of cancer vaccine to inhibit tumour growth. In a multivariate analysis, the presence of Trm in human lung cancer correlated with a better overall survival.

## Results

**Intranasal vaccine administration induces Trm in the lung.** In a first experiment, we showed that the intranasal (i.n.) administration of a vaccine based on a vector targeting dendritic cells (STxB) fused to an E7-derived peptide from HPV16 (STxB-E7) led to the complete protection of mice against an orthotopic head and neck tumour in a therapeutic setting, while only 50% of mice were protected after intramuscular (i.m.) administration of the vaccine (Supplementary Fig. 1). This experiment confirmed our previous results, showing that the mucosal route of vaccine administration was more efficient than the i.m. route to control orthotopic tumour, and that this effect was mediated by CD8[+]T cells[21]. To more comprehensively characterize the differential phenotype of CD8[+]T cells induced by i.n. or i.m. route of immunization, a transcriptomic analysis of sorted E7$_{39–47}$-specific CD8[+]T cells induced by i.n. or i.m. immunization was performed. We found that E7$_{39–47}$-specific CD8[+]T cells from BAL induced after i.n.

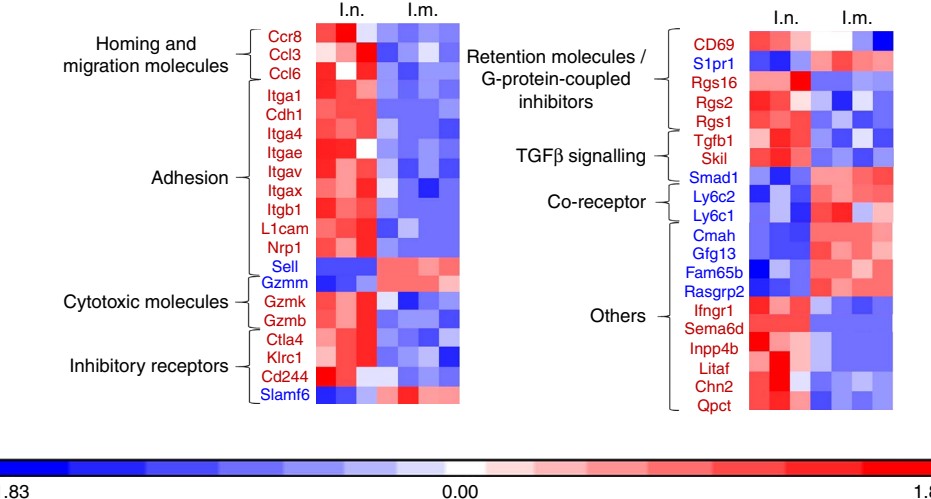

**Figure 1 | Heat map analysis of the gene expression pattern from antigen-specific CD8[+]T cells after i.n. or i.m. immunizations.** Mice were immunized via i.n. or i.m. routes (n = 4 per group) with STxB-E7 + α-GalCer (prime D0) and boosted at D14 with STxB-E7. At day 21, the BALs from the i.n. immunized mice and the spleens from i.m. immunized mice were collected. CD44$^{hi}$Tet[+]CD8[+]T cells were sorted and the RNA was extracted. Whole gene expression microarray analysis was performed according to the procedure described in the methods section. BALs CD8[+]T cells showed a typical Trm gene expression profile, which was not observed in splenic CD8[+]T cells (red square = overexpressed/blue square = underexpressed). Markers in red mean overexpressed in the CD8[+]T cells from the BAL, while markers in blue mean underexpressed in the CD8[+]T cells from the BAL. These extractions were reproduced at least three times.

immunization expressed the core gene defining Trm (Itgae (CD103), Itgb1 (CD49a), Cdh1 (E-cadherine), CD69 and CD244 (2B4)) (Fig. 1). In a comparative analysis with CD8$^+$T cells from the spleen induced after i.m. immunization, the i.n.-induced CD8$^+$T cells also express at higher levels genes encoding adhesion (*Itga4*, *Itgav*, *L1cam*, *Nrp1*), retention (*Rgs16*, *Rgs2*, *Rgs1*), cytotoxic molecules (*Gzmk*, *Gzmb*), and at lower levels the *S1pr1* gene known to favour the migration of T cells to secondary lymphoid organs (Fig. 1). We could not detect E7$_{39-47}$-specific CD8$^+$T cells in the BAL after i.m. immunization to complete the transcriptomic analysis. We confirmed these results at the protein level, as E7$_{39-47}$-specific CD8$^+$T cells expressed CD103 or CD49a or both in the BAL of intranasally immunized mice, while E7$_{39-47}$-specific CD8$^+$T cells in the spleen expressed these two Trm markers at very low levels (<5%) (Fig. 2a). We then showed that the double CD103$^-$ and CD49a$^-$ CD8$^+$T cells in the BAL did not express the CD62L marker, strongly suggesting that they corresponded to effector CD8$^+$T cells (Fig. 2a). Interestingly, all non-effector CD8$^+$T cells in the BAL did not co-express CD103 and CD49a (Fig. 2a). Since CD103$^-$ Trm have recently been reported[25–27] and CD49a was considered as a Trm marker in many studies[2,4,6], we decided to define Trm as cells expressing

CD103 and/or CD49a. A kinetics of these various Trm subpopulations after mucosal immunization showed that at the beginning of the immune response the double-positive CD103 and CD49a CD8$^+$T cells predominate, while at day 30 and day 90 the CD49a$^+$CD103$^-$CD8$^+$T cells were present at higher levels, supporting the use of CD49a as a marker of residency (Supplementary Fig. 2). We then compared the kinetics of induction of Trm and effector CD8$^+$T cells. After i.n. immunization, Trm peaked at D7 (mean 6,000 cells in the BAL) and then declined at day 30, but were still detectable at day 90 (Fig. 2b). Interestingly, the effector CD8$^+$T cells peaked in the BAL at day 7 but at lower levels (mean 500 cells in the BAL) than Trm, and were no more measurable at days 30 and 90 (Fig. 2b). The predominance of the Trm population over the effector CD8$^+$T cells was also confirmed when both populations were measured in the BAL early after the graft of the tumour following a prime-boost i.n. immunization with the STxB-E7 vaccine (Fig. 2c,d).

We then confirmed that Trm cells were also present and persisted in the lung. To discriminate between intravascular CD8$^+$T cells and intraparenchymal CD8$^+$T cells, anti-CD8α mAb was administered intravenously and mice were killed 3 min later to collect the lung, as previously reported (Supplementary

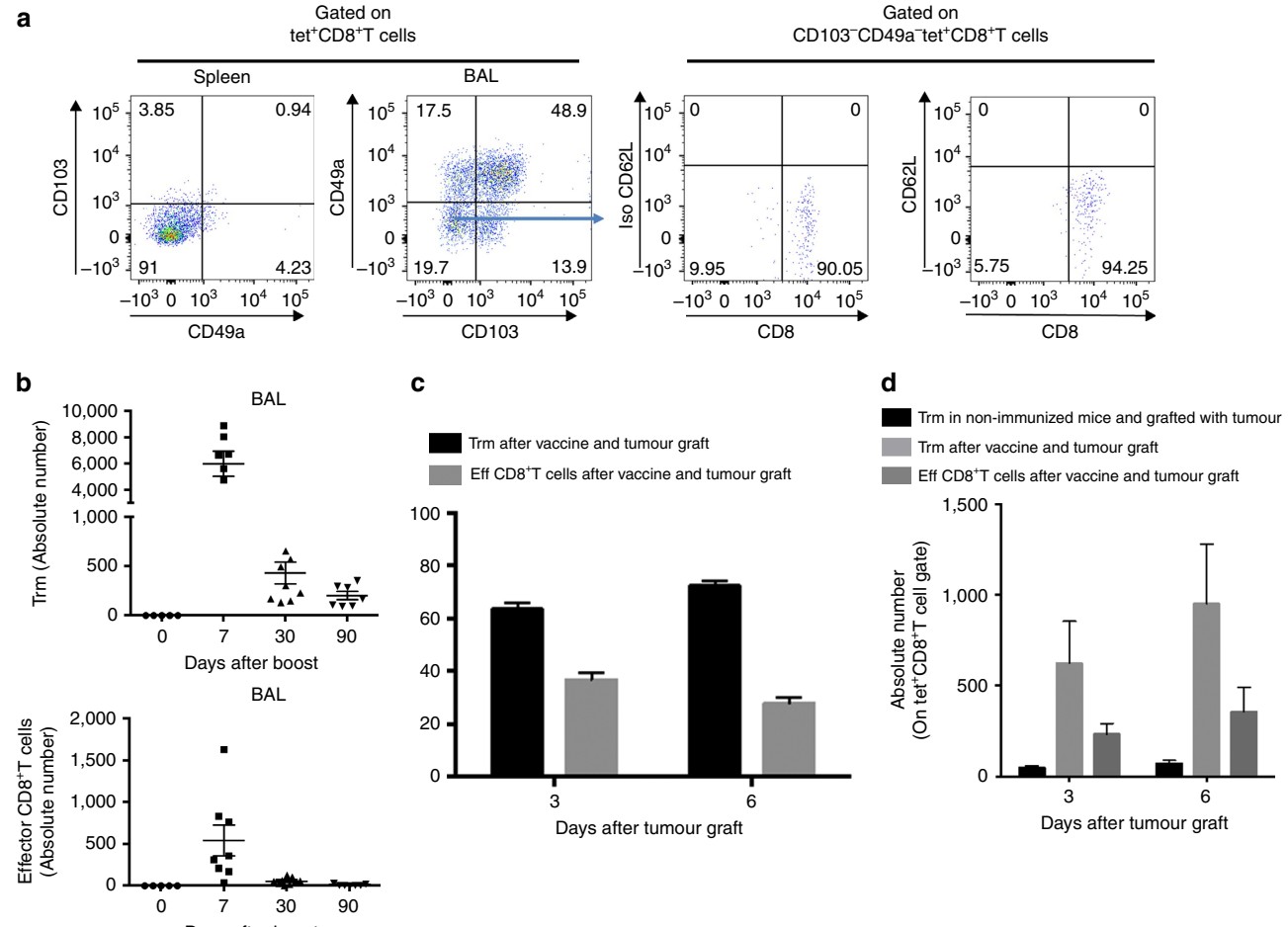

**Figure 2 | Kinetics of Trm and effector CD8$^+$T cells after i.n. immunization.** (**a**) Mice were i.n. immunized (prime (D0)–boost (D14)) with STxB-E7 and BALs and spleens were collected at day 21. Cells were stained and gated on living double-positive E7$_{39-47}$ tetramer and CD8$^+$T cells to assess the expression of CD103 and CD49a. CD49a and CD103 double-negative cells were further stained with anti-CD62L mAb. Isotype controls were included in each experiment. (**b**) Absolute number of Trm cells (CD8$^+$Tet$^+$CD49a$^+$CD103$^-$, CD8$^+$Tet$^+$CD49a$^-$CD103$^+$, CD8$^+$Tet$^+$CD49a$^+$CD103$^+$) and effector CD8$^+$T cells (CD8$^+$Tet$^+$CD49a$^-$CD103$^-$) in BALs measured at D0, D7, D30 and D90 after a vaccine boost. (**c,d**) Mice were i.n. immunized or not with STxB-E7 and 7 days after the boost grafted with the TC1 tumour cells. Three and six days after the graft, the percentage (**c**) and the absolute number (**d**) of Trm and effector CD8$^+$T cells were measured in BALs. Non-immunized mice were also included as controls. Experiments were reproduced at least three times.

Fig. 3A). CD8$\alpha^-$ and CD8$\beta^+$ cells correspond to intraparenchymal CD8$^+$T cells. Using this approach, we demonstrated that although a decay of Trm cells was observed after i.n. immunization, K$^b$-OVA tetramer-positive resident memory CD8$^+$T cells were clearly detected in the lung at least 90 days after vaccine administration (Supplementary Fig. 3B–D).

Trm cells present in the BAL were functional, as they produce IFN$\gamma$ after E7-specific peptide stimulation (Fig. 3). We did not find any difference in the avidity of Trm and effector CD8$^+$T cells derived from the spleen or the BAL (Fig. 3). The cytotoxic potential of these two subpopulations, as assessed by CD107 expression after specific activation of these cells, was not significantly different (Supplementary Fig. 4). However, as previously reported[18], we observed higher levels of PD-1 and Tim-3 expression and PD-1-Tim-3 co-expression on Trm cells compared to conventional effector CD8$^+$T cells (Supplementary Fig. 5).

**Role of Trm in the control of head and neck tumour growth.** In a previous work, we showed that presence of CD8$^+$T cells was required for the efficacy of therapeutic vaccination against orthotopic head and neck tumours[21]. To assess the role of the local specific CD8$^+$T cells in the control of tumour growth, mice were treated or not with FTY720 after their immunization with STxB-E7. FTY720 is a sphingosine-1-phosphate receptor 1 inhibitor that inhibits lymphocyte egress from lymph nodes, thereby preventing the intratumoural influx of systemic effector CD8$^+$T cells after tumour graft[28]. In a preliminary experiment, we showed that mice treated with FTY720 developed a profound lymphopenia as early as 1 day after its administration secondary to the absence of T-cell recirculation (Supplementary Table 3). Fig. 4a showed that half of the mice treated with FTY720 were protected against the growth of tumours that were grafted 7 days after the boost. This experiment strongly argues for a role of local CD8$^+$T cells in the control of tumour growth. To discriminate the role of effector CD8$^+$T cells and resident memory CD8$^+$T cells in this antitumour activity, we built on previous results showing that at D30 after i.n. immunization, only Trm but no effector CD8$^+$T cells could be detected in the BAL of mice (Fig. 2b). We therefore grafted TC1 tumour cells 30 days after STxB-E7 i.n. immunization. We showed that, in this setting, 40% of mice were still protected against tumour growth (Fig. 4b). To increase the protection observed with local Trm, we developed a protocol to increase their frequency in the tongue, before tumour

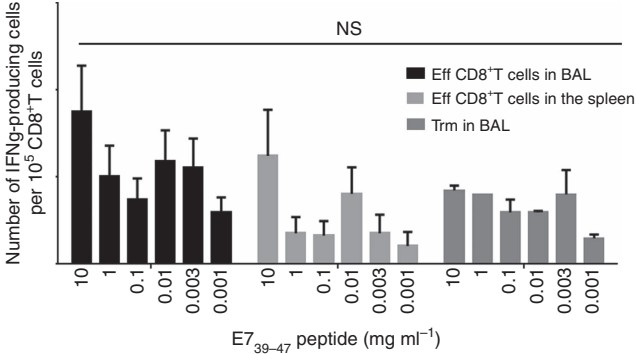

**Figure 3 | Comparative functional analysis of effector CD8$^+$T cells and Trm.** Effector CD8$^+$T cells from BAL and spleen and Trm from BAL were sorted 7 days after the boost i.n. immunization with STxB-E7. Cells were pulsed or not with a set of concentrations of the E7$_{39-47}$ peptide and 24 h later IFN$\gamma$ Elispot was performed. Experiments were reproduced two times and statistical analysis was performed with a two-way analysis of variance (ANOVA). Error bars indicates s.e.m. NS, not significant.

challenge (Fig. 4c). Mice were first immunized via the i.n. route with STxB-E7 at D0 and D14, and then grafted with TC1 at day 21, day 51 and day 111. The number of Trm in the tongue gradually increased after each tumour boost (Fig. 4c). Thirty days after the first immunization, the number of Trm in the tongue was 10, and it increased to 77 or 150 when measured 30 days after the second or third TC1 challenge, respectively (Fig. 4c). At the same time, effector CD8$^+$T cells did not increase following this protocol of repeated tumour challenge (Fig. 4d). All mice were protected and did not develop tumour after the first and second graft with TC1 (day 21 and day 51) (Fig. 4e). In contrast, all non-immunized mice died in less than 30 days after the tumour challenge (Fig. 4d).

To eliminate a possible recruitment of effector CD8$^+$T cells after tumour graft and to demonstrate that tumour protection was really due to Trm, we used the FTY720 molecule. Starting 1 day before the third tumour challenge at D110, mice were treated every day throughout the whole duration of the experiment with FTY720. We demonstrated that, despite the absence of recruitment of peripheral CD8$^+$T cells, more than 80% of mice did not develop tumours (Fig. 4e). The number of Trm present in the tongue correlated with the levels of resistance against tumour growth (Fig. 4a–c).

To support further a role of Trm in the inhibition of tumour growth, we hypothesized that *in vivo* blocking of TGF$\beta$, which plays a critical role in the differentiation of Trm, may reduce their numbers after i.n. vaccination[4,29–31]. Indeed, we found that mice i.n. immunized with STxB-E7 in the presence of neutralizing anti-TGF$\beta$ antibodies exhibited a decrease of E7$_{49-57}$ Trm in their BAL at day 30 after the boost ($P < 0.05$), with no significant modification of effector CD8$^+$T cells (Fig. 5a). In the spleen, a slight increase of specific effector E7$_{49-57}$ CD8$^+$T cells was observed in mice i.n. immunized with STxB-E7 in the presence of anti-TGF$\beta$ (Fig. 5b). We also showed that 5 days after the tumour graft the absolute number of Trm in the tumour decreased in mice in which the vaccine was combined with anti-TGF$\beta$ mAb ($P < 0.05$ by non-parametric Mann–Whitney) (Fig. 5c).When TC1 tumour cells were grafted at day 30 after immunization, more than 60% of mice died when they received the vaccine combined with the anti-TGF$\beta$ mAb, while mice that received the vaccine alone had an overall survival of 60% ($P = 0.049$ by Log-rank test) (Fig. 5d). A decrease of Trm thus has an impact on the efficacy of the cancer vaccine.

To definitively support the role of Trm in the control of tumour growth a parabiosis experiment was done. Mice were i.n. immunized with STxB-E7 and parabiosed 7 days after the vaccine boost. Seven days after parabiosis, spleen and BAL were collected from both the immunized and non-immunized parabionts. We found that E7$_{39-47}$-specific CD8$^+$T cells were present at the same levels in the spleen of the two parabionts (Fig. 6a). In contrast, E$_{39-47}$-specific CD8$^+$T cells with a Trm phenotype were only detected in the immunized parabiont (Fig. 6a). In line with these results, when the TC1 tumour cells were grafted in the non-immunized parabiont, all mice died at day 20, similarly to what is observed in non-immunized naive mice without parabiont grafted with the same tumour (Fig. 6b). In contrast, 25% of the immunized parabiont grafted with the TC1 tumour survived the tumour challenge ($P = 0.0006$ by Log-rank test) (Fig. 6b). The level of protection is lower than in wild-type mice, which may be due to stress or immunosuppression induced by the parabiosis in combination with aggressiveness of this tumour model. In addition, a more conventional parabiosis protocol allowing more time for mice to equilibrate and with separation of parabiosis led to high mortality of mice.

Lastly, when intratumoural concentrations of Trm and effector CD8$^+$T cells were measured after vaccine administration and

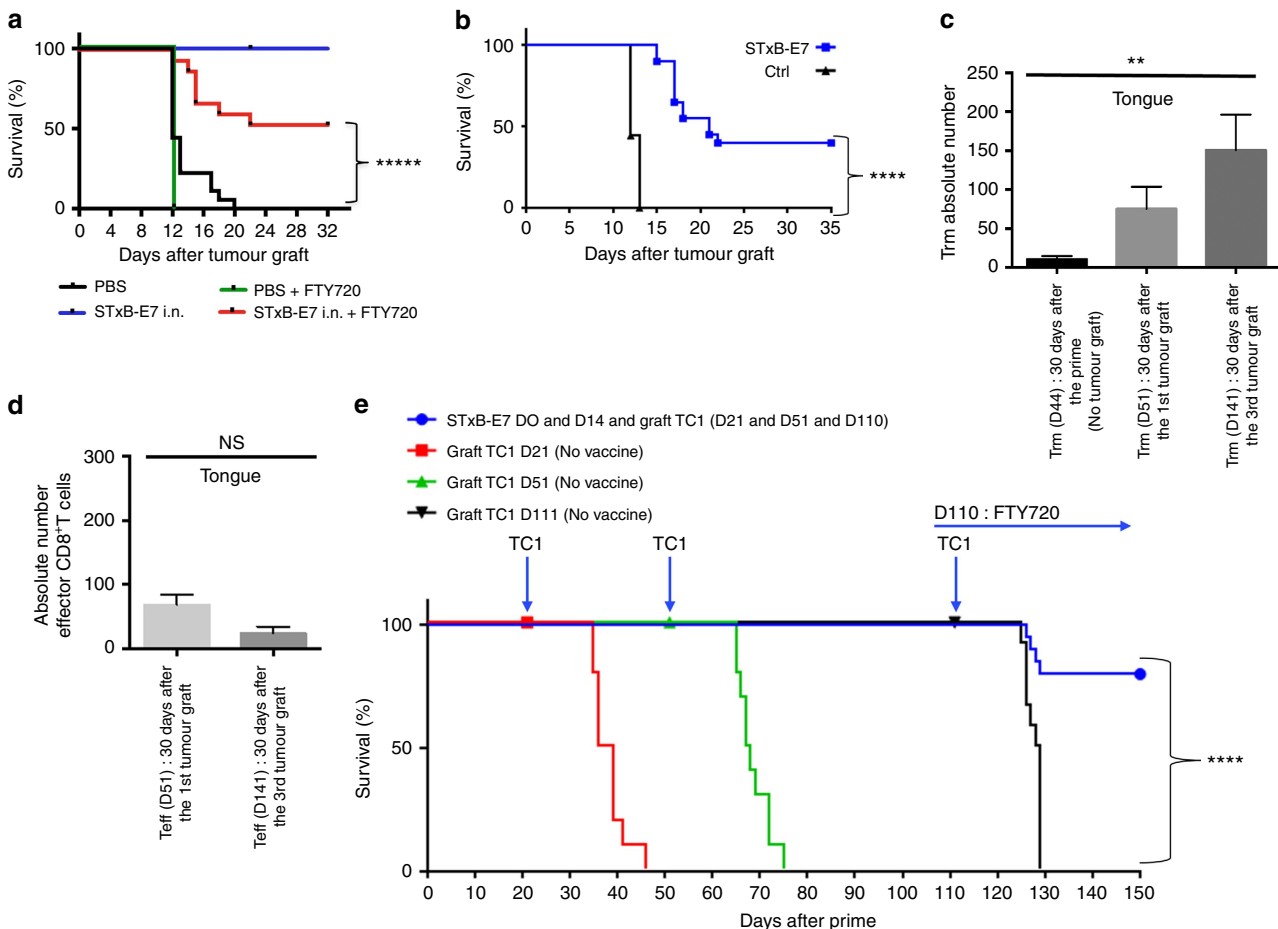

**Figure 4 | Role of Trm in the control of orthotopic tumours.** (**a**) Mice were i.n. immunized or not with STxB-E7 and αGalCer at D0 followed by a boost at D14 with STxB-E7. TC1 cells were grafted at D21. One day before the tumour challenge, the mice were started to be treated or not daily with FTY720 and monitored for survival. (**b**) Mice were i.n. immunized or not with STxB-E7 and αGalCer at D0 followed by a boost at D14 with STxB-E7. One month later, mice were grafted with TC1 cells in the tongue. Survival was monitored and represented by a Kaplan–Meier curve. Experiments were repeated at least three times. ****$P < 0,0001$. (**c,d**) Mice were immunized as in **a**. Trm (**c**) and effector CD8$^+$T cells (**d**) were then measured in the whole tongue 30 days after the boost without TC1 challenge (D44 after priming) for Trm or 30 days after a TC1 submucosae graft at D21 (day 51 after priming) or 30 days after the third TC1 challenge at day 111 (D141 after priming) for both Trm and effector CD8$^+$T cells. One of two representative experiments is shown. (**e**) Mice were not immunized and grafted with TC1 cells at day 21 or day 51 or day 111 (red filled square). Another (green filled triangle) group of mice were (black filled nabla) immunized with STxB-E7 at D0 and D14 and grafted with TC1 at D21, D51 and D111. FTY720 was injected i.p. from (blue filled circle) D110 (one day before the third tumour challenge) until the end of the experiment. Survival was monitored. Experiments were reproduced three times. Statistical analysis for (**a,b,e**) used Log-rank test and for (**c,d**) non-parametric Mann–Whitney test. *****$P < 0.00001$, ****$P < 0.0001$, **$P < 0.01$. Error bars indicates s.e.m. NS, not significant.

tumour graft, the number of intratumoural Trm cells exceeded the number of effector CD8$^+$T cells (Supplementary Fig. 6).

**Trm levels in lung cancer correlated with a good prognosis.** Since the presence of Trm in the BAL of mice was associated with a better survival after vaccination, we tried to extrapolate these results to human cancer patients. For this purpose, we set up an *in situ* double immunofluorescence technique to characterize and measure Trm in a series of 96 human lung cancers. Total CD8$^+$T cells, double-positive CD103$^+$CD8$^+$T cells and double-positive CD49$^+$CD8$^+$T cells were counted (Fig. 7a). For total CD8$^+$T and CD103$^+$CD8$^+$T cells, but not for CD49a$^+$CD8$^+$ (for technical reason), we could dissociate their intratumoural or stromal location thanks to E-cadherin staining of epithelial cells (Fig. 7a). We found that 70% of intratumoural CD8$^+$T cells expressed CD103, a hallmark of Trm, whereas only 41% of CD8$^+$T cells present in the stroma expressed CD103. CD49a was expressed in 26.1% of total CD8$^+$T cells. Interestingly, smokers

who are considered to have more mutations and more prone to generate specific immune responses had an increase of intratumoural CD8$^+$T cells (non-smokers: mean $5.24 \pm$ s.d. 9.26, versus smokers: mean $13.44 \pm$ s.d. 14.9) ($P = 0.03$ by Student's t-test) and CD103$^+$CD8$^+$ Trm cells (non-smokers: mean $3.5 \pm$ s.d. 6.49 versus smokers: mean $10.2 \pm$ s.d. 12.5) ($P = 0.039$ by Student's t-test), while total CD8$^+$T cells and CD103$^+$CD8$^+$T cells were well balanced between the two groups of patients (Supplementary Fig. 3).

In this series of lung cancer patients, the stage ($P = 0.001$ by Log-rank test) and the lymph node status ($P < 0.0001$) were correlated with overall survival. In an univariate analysis with optimized cut-off, both total ($P = 0.035$ by Log-rank test) and intratumoural ($P = 0.004$ by Log-rank test) CD8$^+$T cells, total ($P = 0.006$ by Log-rank test) and intratumoural ($P = 0.001$ by Log-rank test) CD103$^+$CD8$^+$T cells (Trm), and total ($P = 0.009$ by Log-rank test) and intratumoural ($P < 0.001$ by Log-rank test) CD103$^+$ cells correlated with overall survival (Fig. 7b). At 60 months, patients with CD103$^+$CD8$^+$T cells higher than 11 per

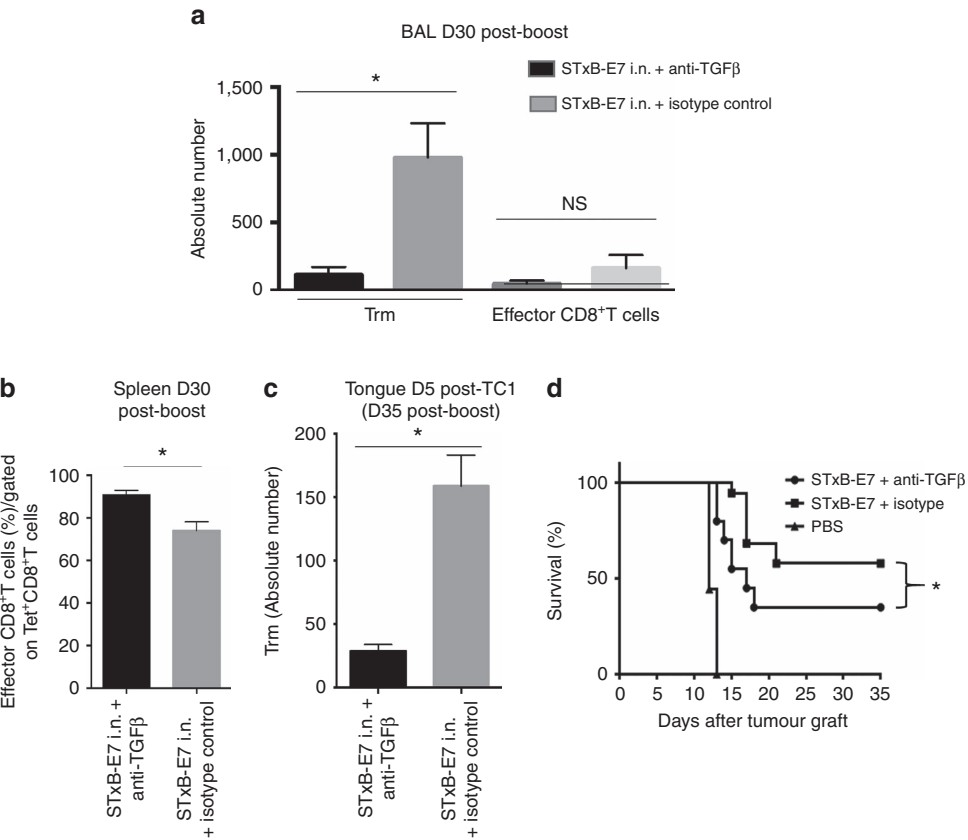

**Figure 5 | Modulation of Trm by TGFβ neutralization decreases cancer vaccine efficacy.** Mice were i.n. immunized (prime-boost) by STxB-E7 and treated in parallel by an anti-TGFβ or its isotype control. Treatment was stopped 1 week after the boost. Trm and effector CD8$^+$T cells were measured 30 days after the boost in the BAL (**a**) and effector CD8$^+$T cells were counted in the spleen 30 days after the boost (**b**). Mice were grafted with TC1 cells in the tongue 30 days after the boost, and 5 days later the number of Trm was measured in the tongue submucosae by flow cytometry to enumerate the absolute number of Trm in the presence or absence of anti-TGFβ (**c**). (**d**) Survival analysis of mice immunized by STxB-E7 combined or not with the anti-TGFβ treatment and grafted 30 days after the boost with TC1 cells. Experiments were performed with at least six mice per immunization group and repeated three times. Statistical analysis for (**a–c**) used non-parametric Mann–Whitney test and for (**d**) a Log-rank test. *$P < 0.05$. Error bars indicates s.e.m. NS, not significant.

field had an overall survival of 84.4%, while patients with a low (<11 per field) CD103$^+$CD8$^+$T cell infiltration had an overall survival of 37.3%. When variables were treated as continuous, intratumoural CD8 (Hazard ratio (HR) = 0.962, 95% confidence interval (CI): 0.931–0.995; $P = 0.02$ by univariate analysis using Cox proportional hazard model), CD103$^+$CD8$^+$T cells (HR = 0.938, 95% CI: 0.892–0.985; $P = 0.011$ by univariate analysis using Cox proportional hazard model) and total CD103$^+$CD8$^+$T cells (HR = 0.973, 95% CI: 0.949–0.997; $P = 0.026$ by univariate analysis using Cox proportional hazard model) correlated with overall survival. Tumour infiltration by CD49a$^+$CD103$^+$T cells did not correlate with overall survival. We also did not observe a relationship between tumour infiltration by these various CD8$^+$T cell subpopulations and progression-free survival.

In a multivariate analysis using Cox proportional hazard model, including both clinical and biological variables, only age older than 70 (HR = 2.079, 95% CI: 1.047–4.128; $P = 0.037$), the pTNM stage (HR = 2.586, 95% CI: 1.451–4.608; $P = 0.001$) and intratumoural CD103$^+$CD8$^+$T cells (HR = 0.264, 95% CI: 0.080–0.873; $P = 0.029$) remain significantly correlated with overall survival (Table 1).

Using the TCGA data, we found a correlation in lung adenocarcinoma between CD8a ($P = 0.028$ using a simple linear regression model) and CD103 gene expression ($P = 0.001$ using a single-linear regression model) and mutational burden (Supplementary Fig. 7), suggesting that these intratumoural CD8$^+$T cells may recognize neoepitopes. We observed a positive correlation between CD8a gene expression and overall survival in lung adenocarcinoma ($P = 0.011$; HR = 1.96, 95% CI: 1.15–3.35) using the Cox regression analysis survival model, but CD103 gene expression was not correlated with clinical outcome in lung cancer regardless of its histology. The absence of impact of CD103 gene expression on clinical outcome could be explained by the fact that the CD103 marker can also be expressed by CD4$^+$T cells either expressing or not expressing the Foxp3 regulatory marker (Supplementary Fig. 8). These data support the value of multiple fluorescence *in situ* analysis, which more accurately defines the subpopulation of infiltrating CD8$^+$T cells associated with clinical outcome.

## Discussion

Using various approaches, we showed that local preexisting CD8$^+$T cells induced after vaccination play a key role in the resistance against the challenge by orthotopic head and neck cancers. Indeed, even after the chemical blockade of the recruitment of peripheral T cells using the FTY720 molecule, a significant control of tumour growth was still observed, which was based mainly upon local CD8$^+$T cells. Parabiosis experiments also showed that local antigen-specific CD8$^+$T cells were observed only in the immunized parabiont, while specific CD8$^+$

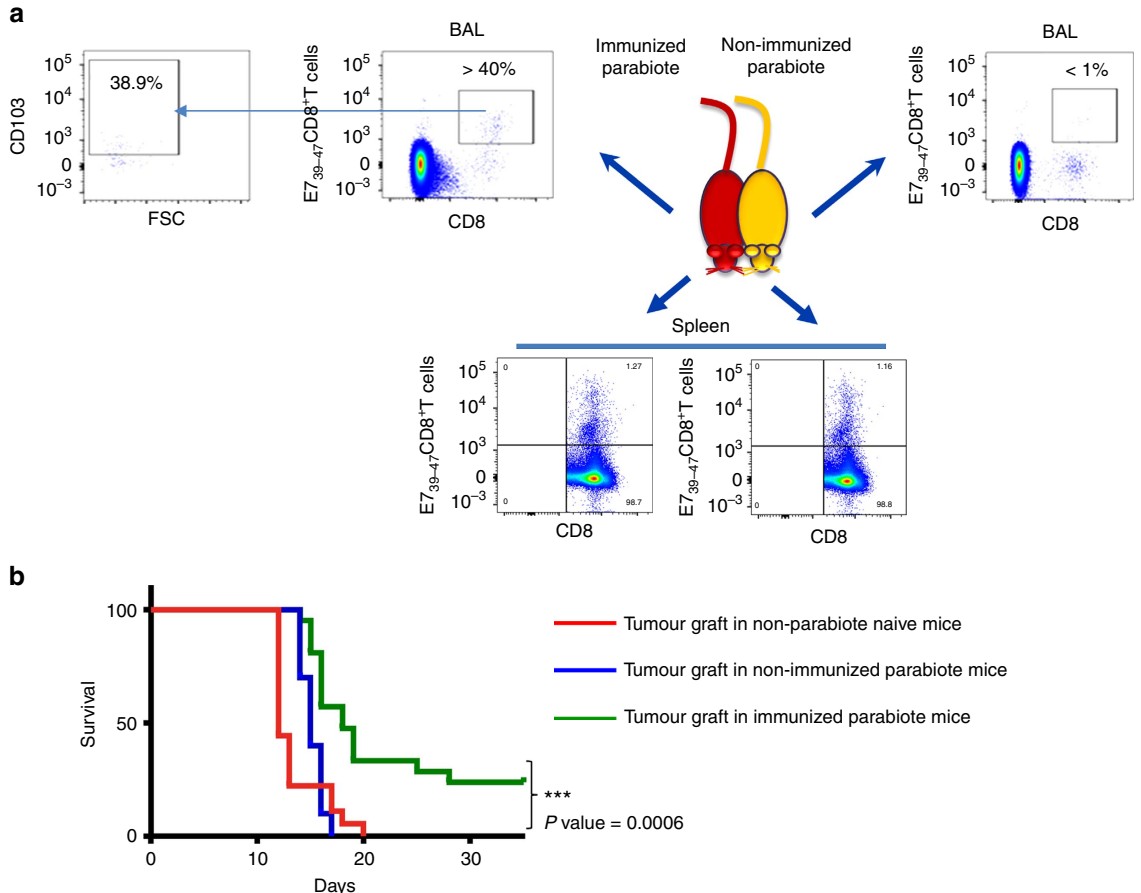

**Figure 6 | Trm controls orthotopic tumour growth as demonstrated in parabiosis experiments.** Mice ($n = 5$) were i.n. immunized with STxB-E7 and 7 days after the boost submitted to parabiosis surgical procedure. (**a**) Dot plot of T cells stained with anti-CD8 and $D^b$-E7$_{39-47}$ tetramer in BALs or spleens from the immunized (left side panels) or non-immunized (right side panels) parabiont mouse 21 days after the surgery. Double-positive cells were gated for CD103 analysis from the BALs of the immunized mice (left). (**b**) Seven days after the parabiosis surgery, the immunized and non-immunized parabionts or naive mice ($n = 5$ per group) were grafted in the tongue with TC1 cells and their survival was monitored. These experiments were independently reproduced three times with similar results. Statistical analysis used a Log-rank test. ***$P < 0.001$.

T cells in the spleen were present at the same levels in both immunized and non-immunized parabionts. Interestingly, only the immunized parabiont was partially protected against a tumour challenge, strongly suggesting that local and not peripheral CD8$^+$T cells were required for this antitumour activity.

The in-depth analysis of the phenotype of the locally induced CD8$^+$T cells showed that early after vaccination two populations coexisted: Trm and CD8$^+$T cells with an effector phenotype. Various arguments converge to show that among these locally induced CD8$^+$T cells, the role of Trm is more important. Indeed, at the peak of the local immune response, concentrations of Trm are 10-fold higher than those of effector CD8$^+$T cells, and only the Trm population persisted locally after 30 days. At this time, when effector CD8$^+$T cells were no more detectable, a protection against tumour challenge was still observed. When we decreased the number of Trm using anti-TGFβ antibodies during vaccination without interfering with the number of peripheral and local CD8$^+$T cells, we also observed a decrease of local tumour control. Conversely, by repeated tumour challenge, which increased the number of persistent local Trm, but not of effector CD8$^+$T cells, we showed that the levels of protection also increased from 40% to more than 80%. The relationship between effector CD8$^+$T cells and Trm cells was not investigated in the present study, but these two cell populations do not appear to differ in terms of cytokine production and avidity and cytotoxic

potential. McMaster et al.[32] reported that Trm possess the ability to produce IFNγ faster than systemic effector memory CD8$^+$T cells, possibly due to the fact that Trm are the first encounter cells after respiratory virus challenge.

To our knowledge this is the first study that demonstrates a key role of Trm for the efficacy of cancer vaccine, as a previous study of Sun et al.[20] reported a correlation between the ability of elicit local Trm and tumour protection. A role of natural tissue-resident innate lymphoid and T cells in the control and adaptive Trm was also recently reported in a preclinical spontaneous tumour model[17]. When dendritic cells were reprogrammed via their treatment with the β-glucan curdlan, a ligand of dectin-1 to induce mucosal antitumour CD8$^+$T cells expressing CD103, these cells were found to be more efficient than control CD8$^+$T cell to inhibit tumour growth[33].

The role of Trm in tumour protection extends previous results about the capacity of these cells to control infectious challenges. In various preclinical studies in mice, the presence of Trm in mouse lungs has been described as a more reliable surrogate marker, than memory-specific T cells in the blood, for reinfection protection[14].

In the present study, vaccination-induced Trm generated a partial tumour protection, when compared to the complete cure of mice when the recruitment of peripheral T cells was not blocked, indicating that both cells were required for optimal antitumour immunity. Via their early cytokine release and

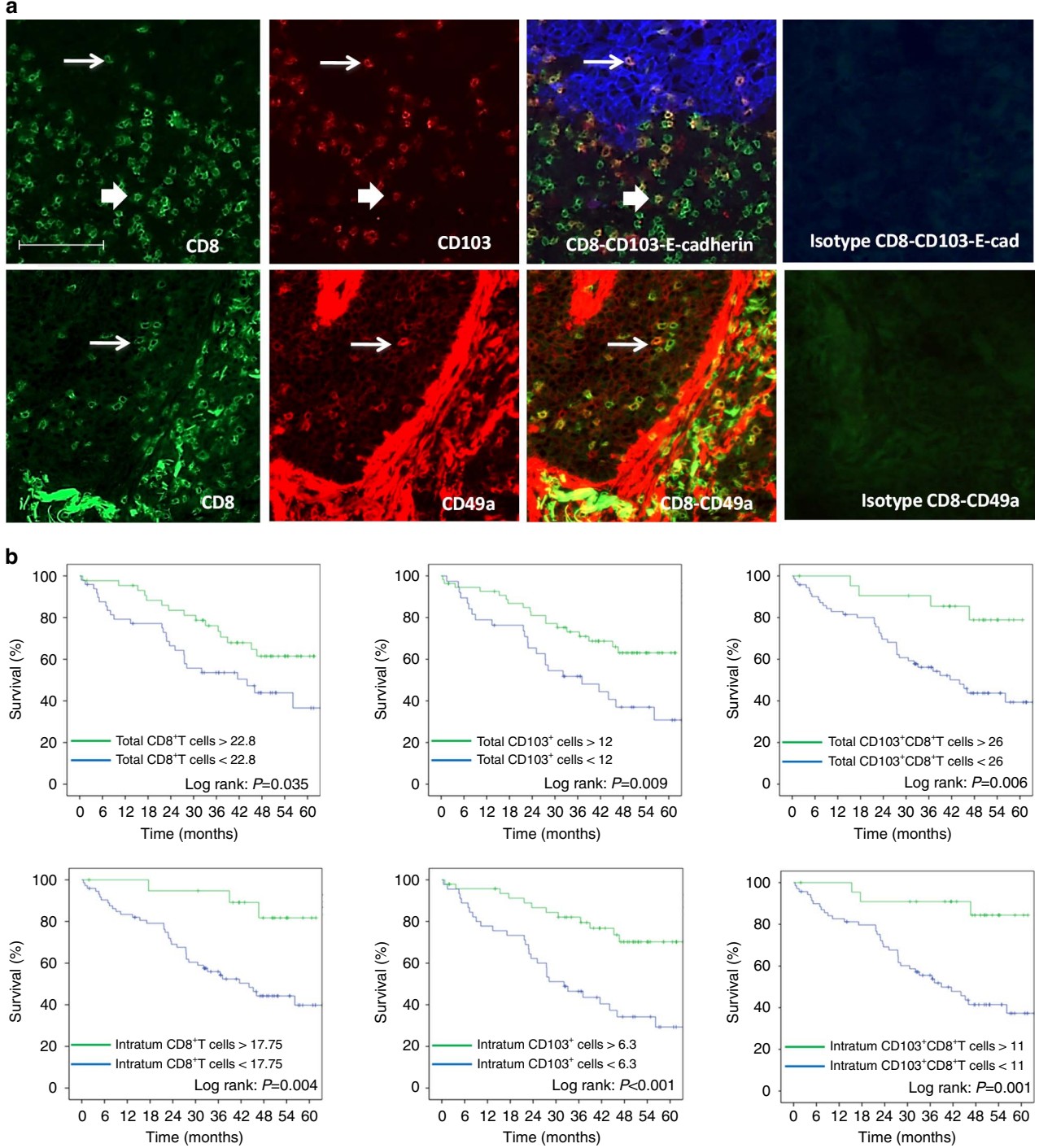

**Figure 7 | Expression and prognostic value of CD103+CD8+T cells and CD49a+CD8+T in lung carcinoma patients.** (**a**) Top: Frozen tissue sections derived from lung carcinoma patients were stained by immunofluorescence with antibodies directed against human CD8 (green), CD103 (red) and E-cadherin (blue). The E-cadherin staining identifies the carcinomatous nests. The colocalization of CD8, CD103 and E-cadherin markers can be detected by merging the mono-staining picture. The thick arrow indicates extra-tumoural CD103+CD8+T cells; the thin arrow indicates the intratumoural CD103+CD8+T cells. Staining with isotypes controls was included for each experiment. (**a**) Bottom: Frozen tissue sections derived from lung carcinoma patients were stained by immunofluorescence with antibodies directed against human CD8 (green), CD49a (red). The colocalization of CD8 and CD49a markers can be detected by merging the mono-staining picture. The arrows indicate the CD49a+CD8+T cells. Staining with isotypes controls was included for each experiment (original magnification × 200). (**b**) Kaplan–Meier analysis of overall survival of patients with lung cancer (*n* = 96). Tumour samples were stained for CD8 and CD103 and E-cadherin to delineate intratumoural or total infiltration (stromal and intratumoural) of cells. Patients were then divided on the basis of CD8+, CD103+ or CD8+CD103+T cell infiltration with optimized cut-off. Statistical analysis used a Log-rank test.

modifications of homing molecule on blood endothelial cells, Trm participate in the secondary recruitment of circulating T cells and other innate immune cells from the blood[8,13,34]. Our results are in line with a recent study reporting that a mucosal *Chlamydia trachomatis* vaccine-induced circulatory and Trm, both of which contributed to protective immunity[10].

**Table 1 | Prognostic factors for overall survival at multivariate analysis using a Cox proportional hazard model.**

| Variable | $\chi^2$ | HR | 95% CI | P value |
|---|---|---|---|---|
| Age >70 years | 3.372 | 2.079 | 1.047–4.128 | 0.037 |
| pTNM stage | 10.384 | 2.586 | 1.451–4.608 | 0.001 |
| Intratumoural CD103$^+$ CD8$^+$ ≥11 cells per field | 4.764 | 0.264 | 0.080–0.873 | 0.029 |

Due to the potential role of Trm in the control of infection and cancer, our results support the use of i.n. mucosal immunization to elicit Trm in head and neck and lung tissue. Mucosal intravaginal immunization also emerges as the most efficient method to elicit Trm in genital tissue[10,19,20]. We have also demonstrated that repeated tumour challenge increased the number of Trm, resulting in a better tumour protection. This result strongly supports the notion that Trm are able to proliferate *in situ* after restimulation as already observed[1,19]. Conversely, we are the first to demonstrate the value of targeting TGFβ *in vivo* in order to modulate the number of Trm.

To assess the relevance of our results in humans, we quantitated the infiltration of Trm in human lung cancer, using a double *in situ* immunofluorescence analysis. We showed that intratumoural Trm defined by the co-expression of CD8 and CD103 correlated with a better overall survival in both univariate and multivariate analysis including intratumoural CD8$^+$ T cells and clinical parameters. A good prognostic value of CD103$^+$ cells has already been reported in early-stage NSCLC and other cancers[35,36]. However, only double staining of CD103$^+$CD8$^+$ T cells, as measured in our study, really corresponds to Trm, as CD103 could also be expressed by NK cells and CD4$^+$ T cells[35]. This bias could explain some controversies in the literature about the impact of Trm on overall survival[18,35]. Wang et al.[20] also reported that intratumoural CD103$^+$ TIL was also an independent prognostic factor for overall survival in a multivariate analysis, thereby supporting our data. This good prognostic value of Trm reinforces the view that they truly represent antitumour T cells. Indeed, we showed that most local vaccine-induced specific T cells had the phenotype of Trm, and that at the steady state they were the only antitumour T cell detected in the tissue. Some of these cells expressed PD-1 and other inhibitory receptors (Fig. 1, Supplementary Fig. 5), as already reported in mice and humans[4,6,18,35]. PD-1 has been shown to be a marker of tumour-reactive TIL in melanoma[37]. As preexisting antitumour T cells are a prerequisite for the clinical success of the blockade of the PD-1-PD-L1 pathway, Trm may represent one target of this therapy and a biomarker of response to this therapy.

This study was performed in one model of tumour-expressing viral protein of HPV, which may constitute a limitation of this study. However, this model has been shown to be predictive of the human efficacy of various cancer vaccines based on DNA or long peptide[38,39]. Another limitation is that the biomarker study on the potential value of TRM has been done on a retrospective cohort, which has to be validated in a prospective study.

Most cancer vaccines failed to demonstrate a clinical benefit in the treatment of cancer, even in the presence of antitumour CD8$^+$ T cells in the blood[40,41]. Conversely, it has been recently reported that Provenge, the only therapeutic vaccine approved in humans, led to local tumour infiltration of CD8$^+$ T cells without the detection of specific CD8$^+$ T cells in the blood[42]. This study and others therefore strongly argue for the induction of persistent Trm as a new goal to be achieved for the development of clinically successful therapeutic cancer vaccines.

## Methods

**Mice.** Six-weeks-old female C57BL/6 (H-2$^b$) (B6) were purchased from Charles River Laboratories. All the mice were kept under biosafety level 2 facility and the

experiments were performed according to European guidelines (EC2010/63) and acceptance by Paris Descartes University ethical committee (CEEA 34): access number project 14-009.

**Cells.** TC1 cells expressing the HPV16 E6-E7 proteins were developed in the laboratory of T.-C. Wu (Department of Pathology, School of Medicine, Johns Hopkins University, Baltimore, MD). Cells were cultured as previously described[21]. Briefly, they were cultured in RPMI 1640 (Life Technologies, ref 7240054 ) supplemented with 10% heat-inactivated fetal calf serum (GE Healthcare, ref A15-101), 1 mM sodium pyruvate (Life Technologies, ref 11 360088), 1 mM non-essential amino acids (Life Technologies, ref 11140035), 100 U ml$^{-1}$ penicillin and 100 µg ml$^{-1}$ streptomycin (Life Technologies, ref 10378016),0.5 mM 2-mercaptoethanol (Life Technologies, ref 31350010), and incubated at 37 °C in 5% CO$_2$. Expression of E7, K$^b$ and cadherin was analysed to characterize these cells. A cell bank for the original cell line was used to make all the experiments and the absence of mycoplasma was tested with the Mycofast test in April 2016.

**Patient cohorts.** A retrospective cohorts of 96 non-treated, non-small-cell lung carcinoma patients who underwent a lobectomy at the thoracic surgery department of European Georges Pompidou Hospital between 2008 and 2010 and for which we have a frozen sample were included in this exploratory study. Patient characteristics are reported in Supplementary Table 1. This study was conducted in accordance with the Declaration of Helsinki and was approved by the local ethics committee (CPP Ile de France no. 2012-05-04). Informed consent was obtained from all participants.

**Reagents.** The adjuvant alpha-galactosylceramide (α-GalCer) was purchased from Funakoshi (Distributed by Euromedex France).

The STxB-E7 vaccine was produced by the chemical coupling of the N-bromoacetylated E7$_{43-57}$ peptide to the sulfhydryl group of a recombinant non-toxic Shiga toxin B-subunit variant according to previously described procedures[21].

**Mice experiments.** Whenever possible during the experiments, the investigators were blinded except for parabiosis experiments and when specific reagents is used in some groups (anti-TGFβ, FTY720, and so on). Each group of mice was matched for age and room for animal house.

**Vaccination protocol.** Mice were immunized as previously described[21]. Briefly, STxB-E7 vaccine (20 µg = 0.5 nmol) was intranasally administered and was associated with α-GalCer (2 µg) at day 0; 14 days later a second immunization was performed with STxB-E7 only (20 µg). A total volume of 25 µl was injected by the i.n. route after anaesthetization.

**FTY720 experiment.** In all, 0.5 mg kg$^{-1}$ per mouse per day of FTY720 (Cayman Chemical) was injected by the intraperitoneal route every day starting 1 day before the tumoural graft. The lymphopenia was checked by whole leukocyte measurement with Hemavet 950FS automate (Drew Scientific).

**TGFβ neutralization.** Mice were treated with anti-TGFβ antibody (BioXcel) or its isotype control by intraperitoneal route. The mice were immunized by STxB-E7 (20 µg) + α-GalCer vaccine (2 µg) and treated with 200 µg of anti-TGFβ the same day, and then treated three times per week with 100 µg of anti-TGFβ each injection. The mice were then immunized a second time by STxB-E7, then treated again with 200 µg of anti-TGFβ and treated three times the following week with 100 µg of anti-TGFβ. In every case, the treatment was stopped the week after the second STxB-E7 immunization.

**Parabiosis.** C57BL/6 mice were immunized or not and the parabiosis surgery was done as previously described[43]. Briefly, mice were shaved along opposite lateral flanks and the skin was disinfected with a veterinary betadine and alcohol preparation. The mice (same age, size and weight) were anaesthetized intraperitoneally (100 mg kg$^{-1}$ ketamine and 10 mg kg$^{-1}$ xylazine) and mirrored incisions were performed at the lateral flanks of both mice that were later connected using surgical clips.

**Flow cytometry.** The $D^b$-E7$_{49-57}$ tetramer was used, according to the manufacturer's recommendations (Beckman Coulter Immunomics) to detect specific CD8+T cells. Briefly, cells were incubated at 4 °C with the PE-labelled tetramer for 35 min. After incubation and washes, labelled anti-CD8 mAbs (eBioscience) were used to phenotype the positive tetramer CD8+T cells. Irrelevant tetramers recognizing a vesicular stomatitis virus (VSV) or LCMV derived peptide $D^b$ molecule were used in each experiment. Naive non-immunized mice were also included as a control for these experiments. To detect tongue tumour infiltrating anti-E7$_{49-57}$/$D^b$-specific CD8+T cells, tumours were mechanically dissociated, filtered, washed in PBS and incubated with the Fc receptor block CD16/CD32 (eBioscience). They were then labelled with the PE tetramer and the anti-CD8 antibody.

For the analysis of the expression of integrin chemokine receptors, inhibitory receptors and CD107, tetramer-positive CD8+T cells were co-stained with anti-mouse CD103 Pacific Blue mAb (5 μg per test, Biolegend, clone 2E7, ref 121418) (Biolegend), anti-mouse CD49a Alexa 647 mAb ((2 μg per test, Becton Dickinson, clone Ha31/8, ref 562113), anti-mouse PD-1 PE-efluo610 (2 μg per test, eBioscience, clone J43, ref 61-9985-82), anti-mouse Tim-3 PE-CY7 (1 μg per test, Biolegend, clone RMT3-23, ref 119716), anti-mouse CD107a and b FITC (5 μg each per test, Becton Dickinson, clone ID4B ref 553793 and ABL-93 ref 558758, respectively). All the cells were labelled using the live/dead cell viability assay (Life Technologies, ref L34966) and analysis performed on gated viable cells.

Analyses were done in a BD Biosciences LSR II or BD Biosciences FACSCalibur with FlowJo (Tree Star) or Cell Quest software (BD Biosciences).

***In situ* immunofluorescence staining.** Tissue samples obtained at the day of lobectomy were frozen and stored at −80 °C. Frozen specimens were sectioned at 4–6 μm with a cryostat, placed on slides, air dried and fixed for 5 min with 100% acetone. Before incubation with antibodies, Fc receptors were blocked with anti-mouse CD16/CD32 (eBioscience) 10% in PBS for 30 min. Two stainings were performed: a triple-staining CD8, 103 and E-cadherin and a double-staining CD8 and CD49a using conjugated antibodies (PE labelled anti-human CD103 (eBioscience); FITC-labelled anti-human CD8 (eBioscience); AF647-labelled anti-human E-Cadherin (Biolegend) and PE-labelled anti-human CD49a (Biolegend)). Isotype-matched antibodies were used as negative controls. Nuclei were highlighted using DAPI mounting medium. The procedure and the source and concentrations of antibodies are detailed in Supplementary Table 2.

Slides of stained lung sections were scanned with an automated fluorescent microscope Axio Scan.Z1 (Zeiss) at the magnification × 200. A coupled Zen software was used to analyse the virtual slides. For each slide, counts were done by a senior pathologist on at least five representative fields with a magnification of × 200.

**Gene expression analysis.** Data were normalized using RMA algorithm in Bioconductor with the custom CDF versus 18 (Nucleic Acid Research 33 (20), e175). Statistical analysis was carried out with the use of Partek GS. All the data have been deposited in NCBI's Gene Expression Omnibus[44] and are accessible through GEO Series accession number GSE77366: http://www.ncbi.nlm.nih.gov/geo/query/acc.cgi?token=ghsrcweibruzvsj&acc=GSE77366.

Fisrt, variations in gene expression were analysed using unsupervised hierarchical clustering and PCA to assess data from technical bias and outlier samples. To find differentially expressed genes, we applied a one-way ANOVA for each gene and made pairwise Tukey's *post hoc* tests between groups. Then, we used *P* values and fold changes to filter and select differentially expressed genes.

DEG enrichment analysis was carried out using Ingenuity (Ingenuity Systems, USA; www.ingenuity.com).

**Statistical analysis.** For mice data, statistical analyses were performed with GraphPad Prism Version 6 software (GraphPad Software Inc.). Data were expressed as means ± s.e.m. and are representative in every experiment of at least three independent experiments (number of experiment indicated in each figure). Significance was assessed with the Mann–Whitney test to compare two different groups and Kaplan–Meier curves to compare the survival of the different mice groups.

For human results, CD8+T cells, CD103+T cells and CD8+CD103+T cells assessed by immunofluorescent staining were firstly analysed as continuous data and then dichotomized using the minimum *P*value approach in order to determine the optimal cut-off for overall survival analysis.

Survival rates were calculated using the Kaplan–Meier method. These survival curves were calculated from the date of diagnosis and all causes of death were considered for survival estimation. Patients were observed until their death or until 25 March 2015, at which time they were censored at the last date they were documented to have been alive. The Log-rank test was used to compare survival curves. Univariate analysis for overall survival was conducted using the Cox proportional hazard model to identify potential prognostic factors of survival. To take into account confounders into survival analysis, a multivariate analysis was performed using a Cox proportional stepwise procedure, including age, sex, pTNM stage, administration of neoadjuvant chemotherapy, administration of adjuvant chemotherapy or radiotherapy and various but not redundant biomarkers that are

associated with the outcome in univariate analysis at a *P* value of <0.100. The 0.1 level was defined for systematic entry into the model. The variation inflation factor and tolerance were calculated to assess multicollinearity before performing multivariate analysis, and multicollinear variables were excluded from multivariate analysis. All tests were two-sided at the 0.05 significance level. All statistical analyses were performed using SPSS version 22.0 (IBM, New York, USA).

**Data availability.** All data have been deposited in NCBI's Gene Expression Omnibus (Edgar *et al.*, 2002) and are accessible through GEO Series accession number GSE77366: http://www.ncbi.nlm.nih.gov/geo/query/acc.cgi?token=ghsrcweibruzvsj&acc=GSE77366.

The LUAD and LUSC data referenced during the study are available in a public repository from the Broad Institute website (https://gdac.broadinstitute.org).

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

## Acknowledgements

This work was supported by a grant from the Institut National du Cancer (INCA) PLBIO IDF (L.J. and E.T.), Ligue contre le Cancer-EL2014 (E.T.), Université Sorbonne Paris Cité (E.T.), ANR-15-CE17-0023-04 (Selectimmunco) (E.T.), Labex Immuno-Oncology (E.T.), SIRIC CARPEM (E.T.) and ERC advanced grant (project 340485) (L.J.). M.N. was a fellow of Fondation ARC and M.O.D. obtained a fellowship from FAPESP.

## Authors contributions

E.T., M.N. and M.O.D. conceived the study and analysed the data. L.J. provided the intellectual guidance and analysed the data. M.N., H.R., M.O.D., T.T., S.K., F.S., M.M., M.T., L.C.S.F., E.M., C.G. and A.G. performed the experiments and analysed the data. T.V., B.R. and J.-B.E. performed the statistical and bio-informatic analysis. E.D. produced the vaccine and analysed the data. E.F., M.R., F.L.P.B., H.R. and L.G. enrolled and classified the patients. C.Bl and C.Ba performed the experiments and analysed the data. E.T. wrote the paper with M.N.'s editing. L.J. provided the intellectual guidance and analysed the data.

## Additional information

**Competing interests:** The authors declare no competing financial interests.

