## [Peer Review File · Nature Communications]

Reviewers' comments:

Reviewer #1 (Remarks to the Author):

The investigators set out to address the role of tissue resident memory (Trm) cells in protection against tumor growth. Using an intranasal vaccination approach, the authors conclude that vaccinal protection against TC1 tumor engraftment is dependent upon the development of Trm cells, but not effector cells, at mucosal sites. By extending these findings to show that the presence of intratumoral Trm cells correlates to a better prognosis in human lung cancer patients, they add significantly to the field of cancer immunology. The author's subdivision of T cells based off of markers of residency (CD103, CD49a) identifies a novel biomarker with promising clinical implications.

The investigators provide striking clinical data that we believe would be bolstered by the inclusion of a similar analysis performed on specimens from the TCGA. In addition, we suggest the authors correlate Trm cell markers with mutational burden using this TCGA data set. The paper's biggest weakness lies in their parabiosis experiment. While their data blocking TGF β and blocking egress with FTY720 create a compelling argument, their parabiosis experiment, the gold standard of the field, is unconvincing. We believe that this results from a flaw in experimental design. The authors should repeat this experiment, priming the mice prior to performing the parabiosis surgery, allowing the mice time to equilibrate, and then separating and providing additional healing time before tumor engraftment is performed. Lastly, we noted that the authors do not provide a temporal characterization of Trm cells in the lung. The decay in the Trm cell population over time has been observed, and as such a description of the Trm composition at the time of actual tumor challenge would better support this work.

Reviewer #2 (Remarks to the Author):

In this manuscript, Dr. Tartour and colleagues first characterize tissue-resident memory T cells (Trm) following mucosal vaccine administration in mice bearing head and neck cancer (HNC). They show that Trm rather than CD8+ Teff memory cells play a major role in mediating antitumor effects of the vaccine and in inhibition of tumor growth. Next, the investigators show that the frequency of Trm cells correlated with a better overall survival in multivariate analysis of human lung cancer.

This is an expertly performed series of in vivo studies of the Trm cell phenotype and functions in a preclinical model of HNC. The demonstration that intranasal vaccine administration induces early (d7) accumulation of antigen-specific CD8+ T cells with the phenotype (CD103+CD49a+) characteristic of Trm is convincingly presented. Further, these cells persist locally after vaccination, are detectable on day 90 and are required for the control of tumor growth (experiments with FTY720 molecule in Fig3). Further, the negative impact of anti-TGF-beta-mAb on the frequency of Trm and also on efficacy of the vaccine confirmed the key role of these cells in vaccine-induced anti-tumor responses. While the tumor control and survival data in the orthotopic model are solid, the mechanisms responsible for the efficiency of Trm cells in mediating anti-tumor responses are not clear. Is this purely due to localization of Trm in the TME and their numerical advantage over convTeff cells? They do not seem to produce more IFN-gamma than Tconv. Are they more cytotoxic? Or are Trms more resistant to tumor-induced suppression? The authors should try and explain why Trm have advantage over antigen-specific CD8+ Teff in executing d more effective anti-tumor responses.

Extension of the Trm studies to in situ phenotyping (CD103+CD8+T cells) in human lung cancer tissues and the reported correlation with patients overall survival in 96 patients as well as the observed impact of smoking on the CD303+CD8+ T cell numbers in the tumor represent provocative but preliminary results. There is some concern that in this retrospective correlative

study of intratumoral CD103+CD8+ T cells vs. CD103+ T cells vs CD8+ T cells there is equally significant OS benefit for all three groups. This result does not support a greater prognostic role of CD103+ Trm cells vs CD8+Teff. Also, the phenotypic characterization of human Trm is not yet in hand and, thus, it is not clear that these investigators are really measuring Trms in human lung cancer tissues. Further, the human component of this paper has nothing to do with vaccination, and perhaps it would be better to omit the human part of the manuscript and focus entirely on vaccine-specific Trms in orthotopic mouse model of HNC.

The manuscript is clearly written and the data presentation is appropriate. The Discussion part is much too long, repeats Results and needs to be abbreviated.

Response to reviewer 1

The investigators provide striking clinical data that we believe would be bolstered by the inclusion of a similar analysis performed on specimens from the TCGA. In addition, we suggest the authors correlate Trm cell markers with mutational burden using this TCGA data set.

Using TCGA data, we investigated a possible correlation between CD103 and CD8a gene expression and mutational burden in lung adenocarcinoma and lung squamous cell carcinoma (LUAD and LUSC datasets). In lung adenocarcinoma, using a linear regression model, we found a weak correlation between mutational burden and CD8a ($p = 0.028$, adjusted $r^2 = 0.025$) and CD103 gene expression ($p\text{-value} = 0.001$, adjusted- $r^2 = 0.058$)(Resp reviewer (RR) Figure 1A and B below)

Figure 1: Correlation between CD8a and CD103 gene expression and mutation burden in the TCGA data set for lung adenocarcinoma.

The mutation burden was computed as the distinct number of non-silent mutations for each patient. Expression levels of CD8a (A) and CD103 (B) were normalized using the z-score transformation. Using a linear regression model, we detected a significant relationship between gene expression and mutation burden.

In contrast, in lung squamous cell carcinoma, we did not find any correlation between mutational burden and CD8a ($p = 0.17$) and CD103 ($p = 0.61$) gene expression

With regards to the relationship between CD8a and CD103 gene expression and clinical outcome, we observed a positive correlation between CD8a gene expression and overall survival ($P = 0.011$; HR = 1.96 - 95% Confidence Interval [1.15,3.35] using the Cox regression analysis survival model in lung adenocarcinoma but not in lung squamous cell carcinoma. CD103 gene expression was not correlated with clinical outcome in lung cancer regardless of its histology.

The absence of correlation between CD103 gene expression and survival may appear to be a contradictory result with respect to our immunofluorescence data, in which univariate analysis showed that total CD103 was correlated with overall survival with an optimized cut-off (main text Fig 6B). However, in contrast to the TCGA analysis, we did not divide our population into two histological forms of lung cancers as expressed in TCGA. In addition, in our previous analysis, when the various parameters were selected as a continuous variable more closely fitting the TCGA analysis, only total ($p = 0.026$) or intratumoral ($p = 0.011$) CD103⁺CD8⁺T cells and not total CD103⁺ cells ($p = 0.219$) were correlated with overall survival. In fact, we found that the CD103 marker could also be expressed by CD4⁺T cells either expressing or not expressing the Foxp3 regulatory marker (RR Fig 2). These data support the value of multiple fluorescence in situ analysis which more accurately defines the subpopulation of infiltrating T cells associated with clinical outcome.

We have mentioned in the text: Page 10 Line 20-25 and Page 11 line 1-5 "Using the TCGA data, we found a correlation in lung adenocarcinoma between CD8a ($p = 0.028$) and CD103 gene expression ($p = 0.001$) and mutational burden (Fig S7), suggesting that these intratumoral CD8⁺T cells may recognize neoepitopes. We observed a positive correlation between CD8a gene expression and overall survival in lung adenocarcinoma ($P = 0.011$; HR = 1.96 - 95% Confidence Interval [1.15,3.35] using the Cox regression analysis survival model, but CD103 gene expression was not correlated with clinical outcome in lung cancer regardless of its histology. The absence of impact of CD103 gene expression on clinical outcome could be explained by the fact that the CD103 marker can also be expressed by CD4⁺T cells either expressing or not expressing the Foxp3 regulatory marker (Fig S8). These data support the value of multiple fluorescence in situ analysis, which more accurately defines the subpopulation of infiltrating CD8⁺T cells associated with clinical outcome".

The paper's biggest weakness lies in their parabiosis experiment. The authors should repeat this experiment, priming the mice prior to performing the parabiosis

surgery, allowing the mice time to equilibrate, and then separating and providing additional healing time before tumor engraftment is performed

We have tried to follow the protocol proposed by the reviewer. However, we obtained a high mortality rate and poorly reproducible results, when a longer interval was observed between parabiosis and tumor graft compared to the previous protocol. We have no explanation for this observation.

We have therefore maintained the previous experiment performed three times with 5 mice per group and we have mentioned, on Page 9 line 3-5, that “a more conventional parabiosis protocol allowing more time for mice to equilibrate and with separation of parabiosis led to high mortality of mice”. However, we could delete this experiment if the reviewer feels that it should not be shown.

Figure 2

Figure 2: CD103 may be expressed by CD4⁺ T cells and regulatory Foxp3⁺CD4⁺ T cells

Frozen tissue sections derived from lung carcinoma patients were stained by immunofluorescence with antibodies directed against CD4 (green), CD103 (red) and Foxp-3 (blue). Colocalization of CD4, CD103 and Foxp-3 markers can be detected by merging the mono-staining pictures. The white arrows indicate co-expression of CD4, CD103 and Foxp3 corresponding to regulatory CD4⁺ T cells and the black arrows indicate co-expression of CD4 and CD103 without Foxp3. Staining with isotype controls was included in each experiment (original magnification x 200).

Lastly, we noted that the authors do not provide a temporal characterization of T_{rm} cells in the lung

We provide a temporal localization of T_{rm} in the lung after vaccination of mice with STxB coupled to ovalbumin (STxB-OVA)(RR Fig 3). To discriminate between intravascular CD8⁺T cells and intraparenchymal CD8⁺T cells, anti-CD8 α mAb was administered intravenously and mice were sacrificed 3 minutes later to collect the lung, as previously reported (Anderson KG et al J Immunol 2012) (RR Figure 3A). After lung dissociation, cells were stained by cytometry. CD8 α -negative and CD8 β -positive cells correspond to intra-lung parenchymal CD8⁺T cells. Using this approach, we demonstrated that although a decay of T_{rm} was observed after intranasal immunization, K^b-OVA tetramer-positive resident memory CD8⁺T cells were clearly detected in the lung at least 90 days after vaccine administration (RR Fig 3B-D). T_{rm} cells predominated over effector T cells at this time (RR Fig 3D).

Thus, using two different vaccines (STxB-OVA and STxB-E7), we detected T_{rm} cells at least 3 months after vaccine administration in the lung parenchyma (RR Fig 3 below) and in bronchoalveolar lavage (main text Fig S2). We have added the following paragraph to the text: Page 6 line 7-14 and we have added a new supplementary figure (Fig S3): “We then confirmed that T_{rm} cells were also present and persisted in the lung. To discriminate between intravascular CD8⁺T cells and intraparenchymal CD8⁺T cells, anti-CD8 α mAb was administered intravenously and mice were sacrificed 3 minutes later to collect the lung, as previously reported (Fig S3A). CD8 α -negative and CD8 β -positive cells correspond to intraparenchymal CD8⁺ T cells Using this approach, we demonstrated that although a decay of T_{rm} cells was observed after intranasal immunization, K^b-OVA tetramer-positive resident memory

CD8⁺T cells were clearly detected in the lung at least 90 days after vaccine administration (Fig S3B-D)".

Figure 3: Persistence of resident memory CD8⁺ T cells in the lung after intranasal vaccine administration

Mice were i.n. immunized (prime (D0) – boost (D14)) with STxB-OVA and the lungs were harvested at D7, D30 and D90 after the vaccine boost. A Left: Representative zebra plot showing gating strategy after intravenous injection of anti-CD8 α to distinguish between lung parenchyma CD8⁺ T cells (CD8 α ⁻ CD8 β ⁺) and vasculature CD8⁺ T cells (CD8 α ⁺ CD8 β ⁺) (Left). A Middle: Frequency of K^b-OVA tetramer-positive CD8⁺T cells gated on live intraparenchymal CD8⁺ T cells. A right: % of resident memory T cells (Trm) in K^b-OVA tetramer-positive intraparenchymal CD8⁺ T cells. B: absolute number of intraparenchymal K^b-OVA tetramer-positive CD8⁺ T cells at various times after immunization. C: frequency of Trm within K^b-OVA tetramer-positive CD8⁺ T cells gated on intraparenchymal CD8⁺ T cells. D: absolute number of TRM or effector T cells within K^b-OVA tetramer-positive CD8⁺T cells gated on intraparenchymal CD8⁺ T cells. Mean \pm sem with 6 to 9 mice/group were represented in each panel of the figure.

The decay in the Trm cell population over time has been observed, and as such a description of the Trm composition at the time of actual tumor challenge would better support this work.

Figure 3c (main text) shows monitoring of Trm in the whole tongue 30 days after the vaccine boost without TC1 challenge (D44 after priming) or 30 days after a TC1 submucosal graft at day 21 (day 51 after priming) or 30 days after the third TC1 challenge at day 111 (D141 after priming). Trm cells clearly increased in the tongue after each tumor challenge. This increase was not observed for effector memory T cells (Main text Fig 3d). Trm cells were also present in the tongue after immunization, while no Trm cells were detected in the tongue in the absence of tumor challenge. These results could explain the better protection observed in Figure 3e (main text) after repeated tumor challenge.

Response to reviewer 2

The mechanisms responsible for the efficiency of Trm cells in mediating anti-tumor responses are not clear. Is this purely due to localization of Trm in the TME and their numerical advantage over convTeff cells? They do not seem to produce more IFN- γ than Tconv. Are they more cytotoxic? Or are Trms more resistant to tumor-induced suppression? The authors should try and explain why Trm have advantage over antigen-specific CD8⁺ Teff in executing a more effective anti-tumor responses.

This is clearly a key point. We did not find any intrinsic difference between resident memory T cells and conventional effector CD8⁺ T cells. Figure 2e (main text) shows that these two types of T cells could not be distinguished on the basis of their avidity and their ability to produce IFN γ . We now provide data concerning the cytotoxic potential of the two populations, which were collected from bronchoalveolar lavage after a prime-boost vaccine administration. When these cells were cocultured with specific antigenic E7 peptide or TC1 tumor cells, the percentage of CD107-positive cells, which reflects degranulation of cytotoxic granules, as well as the MFI of CD107 were not significantly different between resident memory T cells and conventional effector CD8⁺ T cells (RR Fig 4)

Figure 4: Comparative analysis of CD107 expression after activation of effector CD8⁺ T cells and Trm cells with specific antigen.

Mice were immunized with STxB-E7 at D0 and D14 and BAL cells were collected at day 21 and were then transferred to the wells of round-bottom 96-well plates, and incubated for 5h at 37°C with E7₄₉₋₅₇ peptide (10 μ g/ml) or TC-1 cells (10^5 cells/well) in medium containing 1% golgiplug and 1% golgistop and anti-mouse

CD107a FITC (0.1 $\mu\text{g}/\text{well}$) and anti-mouse CD107b FITC (0.5 $\mu\text{g}/\text{well}$). Cells incubated with medium alone served as negative control. After washings, cells were incubated for 5 min with Fc receptor block CD16/CD32, then stained with D^b-E7₄₉₋₅₇ tetramer-PE for 45 min at 4°C followed by staining with anti-CD8, anti-CD103, and anti-CD49a. All cells were labeled using the live/dead cell viability assay and analysis was performed on gated viable cells. Background with medium alone (always < 10%) was subtracted from the results shown.

We also did not observe any significant difference in the expression of proliferative marker Ki67, as both conventional effector CD8⁺ T cells and Trm cells were highly proliferative in the tumor (data not shown).

Tumor-induced suppression was assessed by measuring the expression of PD-1 and Tim-3 inhibitory receptors in these two tumor-infiltrating T cell populations. We observed higher levels of PD-1 and Tim-3 expression and PD-1-Tim-3 co-expression on Trm cells compared to conventional effector CD8⁺ T cells (RR Fig 5). However, the significance of these results has yet to be established, as Trm were not exhausted or dysfunctional, as illustrated by their IFN γ production and CD107 expression (Fig 2e main text, RR Fig 4). Since PD-1 and Tim-3 may also constitute activation markers, these data may suggest a more highly activating state of Trm compared to conventional effector CD8⁺ T cells in the tumor. The preferential expression of PD-1 on resident memory CD8⁺T cells has also been reported in our transcriptomic analysis and by other groups in humans (Webb JR et al Clin Cancer Res 2014)(Djenidi F et al J Immunol 2015). We have therefore added the following paragraph to the text Page 6 line 17-22 and two new figures: “The cytotoxic potential of these two subpopulations, as assessed by CD107 expression after specific activation of these cells, was not significantly different (Fig S4). However, as previously reported (Webb JR et al Clin Cancer Res 2014)(Djenidi F et al J Immunol 2015), we observed higher levels of PD-1 and Tim-3 expression and PD-1-Tim-3 co-expression on Trm cells compared to conventional effector CD8⁺ T cells (Fig S5)”

Figure 5: Preferential expression of PD-1 and Tim-3 on intratumoral E7 specific resident memory CD8⁺T cells

Mice were grafted with TC-1 and vaccinated at D5 and D10 with STxB-E7. Tumors were collected at day 14 and, after cell dissociation, cells were stained with live dead, CD45, CD8, D^b-E7 tetramer, CD103, CD49a, PD-1 and Tim-3. PD-1 and Tim-3 expressions were compared between E7 specific effector CD8⁺ T cells and Trm cells (CD103⁺ and/or CD49a⁺). Results shown are representative of two experiments.

The higher efficiency of Trm cells compared to effector CD8⁺ T cells in mediating antitumor response can probably be explained by the numerical advantage of Trm cells and the preferential localization of Trm cells in contact with the tumor compared to effector CD8⁺ T cells

Indeed, when intratumoral concentrations of Trm and effector CD8⁺T cells were measured after vaccine administration, the number of intratumoral Trm cells exceeded the number of effector CD8⁺ T cells (RR Figure 6). A similar difference was also observed in BAL (main text Fig 2 c and d).

Figure 6: Increased concentration of intratumoral Trm cells compared to effector CD8⁺T cells after vaccine administration

Mice were i.n immunized (prime (D0)-boost (D14)) by STxB-E7 and grafted at day 30 with TC-1 in the tongue. At day 35, the numbers of Trm and effector CD8⁺ T cells were measured in the tongue by flow cytometry. * p < 0.05

In our series of human lung cancers, we showed that 70% of intratumoral CD8⁺ T cells expressed CD103, whereas only 41% of CD8⁺ T cells present in the stroma expressed CD103 (Page 9 line 23-24 and Page 10 line 1). Similar results have also been recently reported in endometrial carcinoma, where CD103⁺ TIL appeared to be exclusively distributed in the tumor nest and not in the stroma (Workel HH et al Eur J Cancer 2016).

We have added these new data to the text: Page 9 Line 12-14: "When intratumoral concentrations of Trm and effector CD8⁺ T cells were measured after vaccine administration and tumor graft, the number of intratumoral Trm cells exceeded the number of effector CD8⁺ T cells (Fig S6)" In the discussion Page 12 Line 10-12: "The relationship between effector CD8⁺T cells and Trm cells was not investigated in the present study, but these two cell populations do not appear to differ in terms of cytokine production and avidity and cytotoxic potential". Page 14 Line 18-19: Some of these cells expressed PD-1 and other inhibitory receptors (Fig 1, Fig S5).

There is some concern that in this retrospective correlative study of intratumoral CD103⁺CD8⁺ T cells vs. CD103⁺ T cells vs CD8⁺ T cells there is equally significant OS benefit for all three groups. This result does not support a greater prognostic role of CD103⁺ Trm cells vs CD8⁺Teff.

I agree that in the Kaplan-Meier analysis of overall survival of patients with lung cancer, both total and intratumoral CD8⁺ T cells, CD103⁺CD8⁺ T cells and CD103⁺ cells correlated with overall survival, as mentioned in the text (Page 10 line 9-12)

However, in a multivariate analysis including both clinical and laboratory variables, only age older than 70 (HR=2.079, 95%CI: 1.047-4.128; p=0.037), pTNM stage (HR=2.586, 95%CI: 1.451 -4.608; p = 0.001) and **intratumoral CD103⁺CD8⁺ T cells** (HR=0.264, 95%CI: 0.080 - 0.873; p = 0.029) remained significantly correlated with overall survival (Table 1) (Page 10 line 21-24). The CD8⁺ T cell population was not identified in this multivariate analysis.

In addition, univariate Cox model analysis showed that the total or intratumoral infiltration of CD103⁺CD8⁺ T cells conferred a better survival advantage than CD8⁺ T cell infiltration with an optimized cut-off for each variable as supported by the Hazard Ratio (HR).

Total CD8⁺ T cells > 22.8 cells/field: HR = 0.517 (95% CI 0.274-0.978): 1.93-fold decrease in the risk of death.

Total CD103⁺CD8⁺T cells > 12 cells/field: HR = 0.261 (95% CI: 0.093-0.736): 3.8-fold decrease in the risk of death.

Intratumoral CD8⁺T cells > 17.75 cells/field: HR = 0.209 (95% CI: 0.064-0.678): 4.78-fold decrease in the risk of death.

Intratumoral CD103⁺CD8⁺T cells > 11 cells/field: HR = 0.178 (95% CI 0.055-0.577): 5.6-fold decrease in the risk of death.

Perhaps it would be better to omit the human part of the manuscript and focus entirely on vaccine-specific Trms in orthotopic mouse model of HNC.

We believe it is useful to report human data on Trm, as:

i) Our results could possibly be extrapolated to humans by demonstrating that the same cells exist in the murine and human tumor microenvironments.

ii) Our data strongly suggest that resident memory T cells (CD103⁺CD8⁺T cells) may provide additional information and prognostic value to tumor-infiltrating CD8⁺T cells alone.

iii) Reviewer 1 asked us to develop this part by TCGA analysis.

The Discussion part is much too long, repeats Results and needs to be abbreviated.

We have deleted two paragraphs in the discussion (Page 12 line 15-19 and Page 13 line 15-21)

REVIEWERS' COMMENTS:

Reviewer #1 (Remarks to the Author):

As a whole, the authors did a good job in addressing our suggestions. The addition of a time line showing the decay of the TRM population in the lung greatly helps their study as it supports the relevancy of the time points they look at. Also, their proposed hypothesis to explain the incongruity between CD103 as a predictive marker using IF vs. TCGA as being due to the addition of different markers identifying CD103 on CD8 T-cells, as opposed to on all cells (and thusly including T-regs) is interesting.

Reviewer #2 (Remarks to the Author):

Thank you for your thorough and informative response to the provided comments. I totally agree with you that the protein-based analysis by multicolor flow cytometry is of greater value for determining expression of CD8a and CD103 in the TME than gene expression profiling. Thank you for elucidating the numerical advantages rMT cells vs conventional effector T cells in respect to vaccination. The revised manuscript reads very well and presents impressive and novel data.